# From Spin Glasses to Negative-Weight Percolation

**DOI:** 10.3390/e21020193

**Published:** 2019-02-18

**Authors:** Alexander K. Hartmann, Oliver Melchert, Christoph Norrenbrock

**Affiliations:** 1Institute of Physics, University of Oldenburg, 26111 Oldenburg, Germany; 2Cluster of Excellence PhoenixD (Photonics, Optics, and Engineering—Innovation Across Disciplines), Welfengarten 1, 30167 Hannover, Germany; 3Institute of Quantum Optics, Leibniz Universität Hannover, Welfengarten 1, 30167 Hannover, Germany

**Keywords:** disordered systems, frustration, phase transition, optimisation, negative weight percolation

## Abstract

Spin glasses are prototypical random systems modelling magnetic alloys. One important way to investigate spin glass models is to study domain walls. For two dimensions, this can be algorithmically understood as the calculation of a shortest path, which allows for negative distances or weights. This led to the creation of the negative weight percolation (NWP) model, which is presented here along with all necessary basics from spin glasses, graph theory and corresponding algorithms. The algorithmic approach involves a mapping to the classical matching problem for graphs. In addition, a summary of results is given, which were obtained during the past decade. This includes the study of percolation transitions in dimension from d=2 up to and beyond the upper critical dimension $d_{u}=6$, also for random graphs. It is shown that NWP is in a different universality class than standard percolation. Furthermore, the question of whether NWP exhibits properties of Stochastic–Loewner Evolution is addressed and recent results for directed NWP are presented.

This review starts by discussing spin glasses, which are one of the most fundamental types of of disordered (magnetic) systems exhibiting frustration. It is shown that, for planar spin glasses, the ground state can be computed using algorithms which find suitably defined minimum weighted paths of interaction constants. These concepts can be generalised to arbitrary dimensions. This leads to the *negative-weight percolation problem* (NWP), which can be defined and studied on arbitrary graphs, i.e., in arbitrary dimensions. Thus, in Section 2, a quick introduction to graph theory is presented. The study of path-like objects in statistical physics is not new (although restricted to settings without the special properties arising from the negative weights), with some examples given in Section 3. The precise definition of NWP is given in Section 4. In addition, the different mappings for undirected and directed NWP are detailed, allowing one to numerically solve large NWP instances using fast matching algorithms. In Section 5, an overview of reported results for NWP is given. In particular, the percolation transition of the undirected NWP is discussed for NWP on lattices in dimensions 2 to 7. It is also shown that the upper critical dimension is $d_{u}=6$, accompanied by results for mean-field NWP. Furthermore, results for the distribution of loops and for diluted NWP are shown. It is also shown that NWP is *not* a process belonging to the class of *Stochastic–Loewner Evolution*, also called *Schramm–Loewner Evolution* (SLE) [1]. Finally, the main results for *directed* NWP are mentioned. This review closes with conclusions.

## 1. Spin Glasses, Ground States and Domain Walls

Spin glasses [2,3,4,5,6] are disordered magnetic alloys, which exhibit many puzzling properties, in particular at low temperatures. Despite more than 10,000 scientific publications devoted to spin glasses, many properties are still not well understood. Furthermore, the study of these systems by analytical and numerical means has led to many cross-disciplinary developments and other applications, like error-correcting codes [7], neural networks [8], or optimisation problems [9].

Experimentally, many spin-glass materials exist. For example, consider a three-dimensional system of a conducting non-magnetic material like gold and randomly exchange some fraction *x* of the gold by a magnetic material like iron (Fe${}_{x}$Au${}_{1-x}$). To observe the specific spin-glass behaviour experimentally, the alloy is put into a weak magnetic field with strength *h*. Then, the global magnetisation *m* is measured, from which one obtains the standard magnetic susceptibility $\chi ={\left.\frac{\partial m}{\partial h}\right|}_{h\to 0}$. When recording χ while varying the temperature *T*, a peak at a very low temperature $T_{G}$ is visible. The value $T_{G}$ is typically in the range of a few 10 Kelvin. This may change depending on the way the sample is prepared and on the concentration of the magnetic compound. This peak indicates some kind of phase transition. However, in contrast to standard phase transitions, like for a classical ferromagnet, the susceptibility shows just a kind of cusp, but not a real divergence. This shows that the phase transition is somehow unusual. Nevertheless, for spin glasses, a divergence can also be found when looking at the nonlinear susceptibility $\chi_{nl}$ [6]. This susceptibility is also obtained by putting the system into a magnetic field (strength *h*) while measuring the resulting magnetisation and but now comparing (or fitting) with an extended expression $M\left(h\right)=\chi_{l}h+\chi_{nl}h^{3}+O\left(h^{5}\right)$. To observe magnetic order in experiments, neutron scattering is typically used. Nevertheless, for spin glasses, one observes no spatial (e.g., ferro- or antiferromagnetic) order of the spin orientations below the transition temperature $T_{G}$. However, in the equilibrium *spin-glass phase*, there exists some kind of ordering. It is characterised by the magnetic moments being frozen to certain orientations relative to each other, which becomes visible by significantly conserved correlations between neighbouring spins.

Even more puzzling is the non-equilibrium behaviour, e.g., *ageing* experiments. Here, the time evolution of spin glasses are investigated, while varying the initial conditions, i.e., the history, which seems to be memorised even after a long time. This important field is beyond the scope of this introduction section; please refer to specialised reviews [10].

This strange behaviour is caused by the special types of interaction found in spin glasses. For example, for the Fe${}_{x}$Au${}_{1-x}$ alloy, the indirect-exchange interaction, called *RKKY* interaction, is present (named after the researchers Ruderman, Kittel, Kasuya, and Yosida, who proposed it). For a magnetic spin $S_{i}$, which is placed in a cloud of conducting electrons, the electrons are polarised by the magnetic spins. As RKKY found out, this polarisation oscillates in sign and decreases in magnitude with distance. Another spin $S_{j}$ which is located at some distance *r* from $S_{i}$ will therefore lead to an effective interaction between this pair of spins with an energy E(r), which oscillates and decreases as a function of *r*; see Figure 1.

In the alloy of iron and gold, the magnetic atoms are placed randomly in the gold matrix. Thus, each spin of an iron atom interacts with any other (close) spin, depending on the distance, either ferromagnetically or antiferromagnetically. Therefore, there will be pairs of spins which prefer energetically to be aligned in parallel while others favour a non-parallel relative orientation. At temperatures below the transition temperature, these mixed interactions lead to a frozen non-regular pattern of relative spin orientations. This makes it comprehensible why within neutron–scattering experiments no spatial order can be found.

Note that there are other physical systems which exhibit other types of interactions leading to a similar behaviour. Anyway, the main ingredients which yield the typical spin–glass behaviour are *mixed signs of interactions* and *disorder*. Inevitably, there will be spins which cannot fulfil all constraints imposed by the bonds connecting them to their neighbours. Hence, some antiferromagnetic bonds will connect parallel spins and the other way around. This is usually summarised by saying that it is impossible to *satisfy* all bonds. The impossibility of satisfying all bonds is called *frustration*, a concept which was coined by Toulouse [11]. A simple example of a frustrated system is shown in Figure 2.

To build a theoretical model of a system, one would like to have as few ingredients as possible in order to observe the desired behaviour. For a spin glass, it means that one would like to avoid the complicated shape of the RKKY interactions and also the random placements of the spins. This is achieved by the *Edwards–Anderson* (*EA*) *model* [12], which consist of *N* spins arranged on a *regular* lattice, e.g., a quadratic ($N=L^{2}$) or cubic ($N=L^{3}$) one, where *L* denotes the linear extension. In the EA model systems with a high magnetic anisotropy are described. This means the spin vectors orient preferentially along one distinguished axis. Thus, the model exhibits so-called *Ising spins*, where in normalised units the variables take $\sigma_{i}=\pm 1$. The EA model has been analytically and numerically studied extensively since its invention. It has been found that it describes the most important spin-glass phenomena, like the low-temperature spin-glass phase, the unusual phase transition, and the non-equilibrium behaviour.

Next, we present the EA model in detail. Only interactions between nearest neighbours of spins are considered. The randomness is included in the model not by a random placement of the spins but by including random *interactions*, which are either ferromagnetic or antiferromagnetic. The underlying Hamiltonian reads
(1)$$H\equiv -\sum_{\langle i,j\rangle }J_{ij}\sigma_{i}\sigma_{j}-B\sum_{i}\sigma_{i},$$
wherein the sum 〈i,j〉 considers all nearest-neighbour pairs of spins. The bonds connecting spins *i* and *j* are given by values $J_{ij}$. The second term describes the presence of an external field *B*. For this work, we only consider B=0. This absence of a magnetic field leads to a spin-flip symmetry of the Hamiltonian (Equation 1). This means, for each configuration $\left\{\sigma_{i}\right\}$, the “reversed” configuration $\left\{-\sigma_{i}\right\}$ exhibits the same energy.

A set of bond values determines a so-called *realisation* of the system. The values of the bonds do not change during a simulation or calculation—one calls this *quenched disorder*. For each realisation, the bond values $J_{ij}$ are randomly drawn from a given probability distribution. Very common are the ±J (bimodal) distribution and the Gaussian distribution. An example for a square realisation is displayed in Figure 3. The behaviour of each realisation depends on the actual choice of bond values, even a full ferromagnetic realisation is part of the disorder ensemble, although it is very unlikely. In any case, to obtain meaningful statements about the ensemble, one has to perform an average over the realisations. For simulations, one typically considers hundreds or even thousands of independent realisations.

Since we want to investigate the low-temperature behaviour of spin glasses, one useful approach is to study the *ground states* (GSs), i.e., the configurations of lowest energy. Here, we now concentrate on two dimensions, more precisely, on planar interaction patterns. This means, it is possible to draw the systems of bonds such that no intersections occur. For planar systems, there is a sophisticated approach [13,14], which allows one to calculate GSs quickly, i.e., in polynomial time. This means large systems can be addressed.

To understand the approach to obtain ground states, we start by looking at the system where all spins are set, e.g., to “up” $\sigma_{i}=+1$. As a result, the ferromagnetic bonds are all satisfied, while all antiferromagnetic bonds are not satisfied; see Figure 4. The basic idea is to look for closed domain walls with positive energy (“weight”). Domain walls correspond to paths in the *dual lattice*, where the sites of the dual lattice are the elementary 4-spin plaquettes of the lattice. The sites of the dual lattice are shown by filled circle symbols in Figure 4. The energy of a path in the dual lattice is equal to the sum of the energetic states of the bonds crossed by the path. Thus, each unsatisfied bond $J_{ij}$ contributes energy $|J_{ij}|$ while each satisfied bond contributes energy $-|J_{ij}|$. Hence, if a closed domain wall with positive energy exists (shown by dashed lines and grey areas in Figure 4), then one can flip all the spins ($\sigma_{i}=+1↔-1$) which are inside the domain walls. This means that the energetic state of each bond crossed by the domain wall is reversed, hence the total energy contributions of the domain wall become negative. This results in an overall energy decrease.

Consequently, if no more domain walls with positive energy are found, it is clear that one has obtained a GS for the system. The ground state for the sample system of Figure 3 is shown in Figure 5.

Furthermore, often one is interested in forcing domain walls into GS configurations. The scaling of the domain-wall energy often follows [15] a power law $L^{\theta }$ as a function of the lateral system size *L* and allows one to draw conclusions on the properties of the spin glass low-temperature phase (For example, if θ>0, the spin glass phase is stable against small thermal fluctuations. Conversely, for a negative value of θ, no ordered phase can be stable at any temperature, which means the critical temperature is $T_{c}=0$.). These domain walls can be created as follows: after obtaining the ground state, say for a planar spin glass, we consider the spins in the first and in the last column. Next, we introduce additional bonds between the spins of these two columns, row by row, i.e., creating periodic boundary conditions. These bonds will be chosen with a sign which is *not* compatible with the GS orientation of the spins; see Figure 6. Furthermore, the absolute value of these bonds shall be extremely big. When the GS of the modified system is calculated, it will enforce that the spins of the first and last column flip their relative orientations in comparison to the ground state. For example, the spins in the left column will remain as they are, while the spins of the right column flip with respect to the GS. Nevertheless, there must be another line within the system, which separates the spins which remain in their GS (“left”) and which flip (“right”). Clearly, since it is a GS of the modified system, this domain wall with have a minimum energy. Since this domain wall (or path on the dual lattice) runs from the top of the system to the bottom, it is not closed, i.e., it is not a cycle.

Thus, open and closed closed paths of minimum or maximum energy are useful when calculating GSs and domain walls of planar spin glasses. In the following sections, we will see how such paths can be actually obtained by an algorithm in a running-time which grows only polynomially with system size. Hence, the calculation is fast opposed to all known algorithms for “hard” problems. These algorithms exhibit an exponential growth. By using this approach, large planar spin glasses of up to $3000^{2}$ spins can be treated [14]. In general, the application of this and similar fast GS algorithms [16] have been successfully used to clarify the nature of GSs of two-dimensional spin glasses [13,14,17,18,19,20,21,22,23,24,25,26,27,28,29,30,31,32,33,34].

Now, one can generalise the spin-glass ground state and domain-wall problems: Given a set of positive and negative “weights” (corresponding to the energies) in a (dual) grid or any other lattice, can one find a collection of cycles and possible open paths (with given starting and end points) such that the total weight, which is defined as the sum of the weights located along a path, is minimised or maximised? Note that, since the minimisation of the weights corresponds to the maximisation of the negated weights, we can focus on one of these cases, which will be the minimisation here. In particular, one can ask this question for lattices of any dimension. For dimensions higher than two, these paths do not correspond to domain walls any more. Nevertheless, the paths can be of interest on their own. In particular, one can ask whether such paths, depending on the total concentration of the negative and positive weights, tend to be small or rather large and system spanning, i.e., whether they *percolate*. This leads to the *negative-weight percolation problem*, which will be defined more precisely below. Before we come to this, we need some concepts from graph theory.

## 2. Graphs

In this section, we will give a short introduction to graph theory. This will allow us to explain the fundamental graph matching problem on which the calculation of spin-glass ground states is based and all necessary algorithms used here.

Graphs represent objects with relations, interactions or connections between them. They are ubiquitous. Examples are cities connected by train lines, computers connected by Internet connections, people connected by social relations like friendships or proteins which are considered to be “connected” if they physically interact with each other. Note that these relations have no orientation, e.g., a train, in principle, can go from London to Birmingham, or vice versa. There are also relations where the direction plays a role. Examples are web pages connected by URL links, animals in food webs which are connected if animal A eats animal B, or scientific papers connected by citations. Formally, graphs are defined as follows:

**Definition** **1.**
*An *undirected* graph G=(V,E) consists of a set V of *vertices* or *nodes* and a set $E\subset V^{\left(2\right)}$*edges* or *arcs*e={i,j}∈E which connect nodes pairwise. $V^{\left(2\right)}$ denotes the set of all two-element subsets of V, i.e., the set of all possible edges. Note that the edges {i,j} and {j,i} are the same.*

*For a *directed* graph G=(V,E), the edges have an orientation: E is now a set of ordered pairs (i,j)∈V×V. Thus, (i,j) denotes an edge from vertex i to vertex j which is different from the edge (j,i) which runs from j to i.*


For all applications, it is sufficient to consider only *finite* graphs. These are the graphs where the set of nodes is finite (and thus also the set of edges). Next, we introduce more useful notations. We start with undirected graphs. For any edge {i,j}∈E, the node *j* is called a *neighbour* of *i* (and *i* a neighbour of *j*). Any nodes connected by an edge are *adjacent*. The set N(i)={j|{i,j}∈E} denotes all neighbours of node *i*. The number of neighbours of node *i* is called the *degree*
d(i). Therefore, it is the cardinality of the neighbour set: d(i)=|N(i)|. A vertex with no neighbours, i.e., d(i)=0, is called *isolated*. $G=(V,V^{\left(2\right)})$ is called the *complete graph*, i.e., each vertex is connected to all other vertices.

A sequence of vertices $v_{1},v_{2},\ldots ,v_{k}$, where neighbouring nodes are connected by edges: $\{v_{i},v_{i+1}\}\in E\;\;\forall i=1,2,\ldots ,k-1$, is called *path* from $v_{1}$ to $v_{k}$. The number of edges along the path is called the *length* of the path, i.e., k−1. A path is called *closed* if $v_{1}=v_{k}$. If all nodes on a closed path appear only once, except $v_{1}=v_{k}$, we call it a *cycle*. Whenever a graph can be drawn on a sheet of paper with no two edges intersecting, it is denoted as a *planar graph*.

**Example** **1.**
*In Figure 7, the undirected graph G=({1,2,3,4,5,6},{{1,3},{1,4},{1,6},{2,4},{3,4}}) is displayed. The nodes are shown as circles and the edges as straight lines connecting the nodes. The number of vertices is 6 and the graph contains 5 edges. For example, the nodes 2 and 4 are adjacent. Vertex 4 has the set of neighbours N(4)={1,2,3}. Thus, node 4 has degree 3. Node 2 has only one neighbour, i.e., d(1)=1. Node 5 is isolated; it has no neighbours. The graph contains the path 3,4,1,6 from node 3 to node 6. The path has the length 3. There is also one cycle 3,4,1,3.*


For directed graphs, most definitions carry over. Since the edges (i,j) exhibit an orientation, one says they are *outgoing* from node *i* and *incoming* to node *j*. The number of incoming edges is denoted by the *in-degree*
$d_{-}\left(k\right)=\left|\left\{i|\left(i,k\right)\in E\right\}\right|$, and the corresponding *out-degree* is $d_{+}\left(k\right)=\left|\left\{j|\left(k,j\right)\in E\right\}\right|$. A sequence of vertices $v_{1},v_{2},\ldots ,v_{k}$ constitutes a *directed path* from $v_{1}$ to $v_{k}$ if the nodes are connected by correctly oriented directed edges: $(v_{i},v_{i+1})\in E\;\;\forall i=1,2,\ldots ,k-1$. Thus, if one imagines walking along the path, one always has to follow the direction of the edges.

**Example** **2.**
*In Figure 8, the directed graph G=({1,2,3,4,5,6},{(1,3),(1,4),(4,2),(4,3),(6,1)}) is shown. Node 1 has out-degree $d_{+}\left(1\right)=\left|\left\{3,4\right\}\right|=2$ and in-degree $d_{-}\left(1\right)=\left|\left\{6\right\}\right|=1$. The graph contains the directed path 6,1,4,3 starting at node 6, via nodes 1 and 4, to node 3. It has length 3. There are no cycles.*


Values are often associated to the edges. For example, if the nodes represent cities and the links are train lines, one could associate the distance (in km) or the (typical) travel time (in h) to the edges. This can be put in a definition as follows:

**Definition** **2.**
*A graph G=(V,E,ω) with edge weights ω:E→R is called a *weighted graph*.*


This definition of weights for edges can be extended in a natural way:

**Definition** **3.**
*Let L⊂E be a subset of edges. The weight of L is defined as the sum of the weights of the edges contained in L:*
(2)$$w\left(L\right)=\sum_{e\in L}w\left(e\right)\;.$$


In the case that the edge weights are, e.g., distances, the definition of the path length is modified. Let a path $v_{1},\ldots ,v_{k}$ be connected by the edges $L=\{\left\{v_{i},v_{i+1}\right\}|i=1,\ldots ,k-1\}$. The length of the path is now given by w(L) instead of just the number of edges. The *shortest path problem* is to obtain among all paths connecting two given vertices *i* and *j* that path which exhibits the shortest length. For all weights being positive, i.e., corresponding to distances, this is a classic graph-theoretical algorithmic problem. It can be solved in polynomial time by applying, e.g., the Dijkstra algorithm [35]. Nevertheless, for NWP, negative weights are also allowed, leading even to cycles of negative weight. In this case, standard shortest-path algorithms do not work. In particular, shifting all edge weights into the regime of positive weights is not equivalent, as it is shown below in Section 4.2.

As we will see in Section 4, in order to treat the NWP, we have essentially to obtain a *minimum-weight perfect matching*, but not for the original graph *G*, instead for a suitably constructed auxiliary graph $G_{A}$. Below, we will explain step by step how $G_{A}$ is constructed, but first we introduce matchings in general.

**Definition** **4.**
*Let G=(V,E) be an undirected graph. A matching M⊂E is given by a subset of edges, with the condition that no pair of edges in M is incident to the same vertex [36,37]. Therefore, for all vertices i of the graph, it holds that i∈e for at most one edge of the matching.*


All edges which are part of a matching are denoted as *matched*, while edges which are not part of the matching are called *free*. In the same way, each vertex is called *matched* if it belongs to edge e∈M. Vertices not being matched are called *free* or *exposed*. Two nodes *i* and *j* are called *mates* if they are matched.

A matching *M* which exhibits a maximum number of edges is denoted as a *maximum-cardinality matching*. If all nodes of a graph are matched, then the matching is a *perfect* matching. Thus, automatically it is of maximum cardinality. On the other hand, for some graphs, the maximum-cardinality matching is not perfect, i.e., no perfect matching exists. The other way around, the empty set is always the matching of minimum cardinality.

**Example** **3.**
*Matching*

*An example for matching in graphs is shown in Figure 9. The edges which are part of the matching and the matched nodes are marked by bold lines. The matching M={{1,4},{2,5}} is shown on the left. Here, e.g., edge {2,5} is matched, while edge {2,6} is free. The nodes 1, 2, 4 and 5 are matched; on the other hand, nodes 3 and 6 are free.*

*In the second panel of this figure, the perfect matching M={{1,5},{2,6},{3,4}} is displayed. Thus, there is no exposed vertex.*


For a given weighted graph G=(V,E,w), we can extend the definition to *weighted matchings*. Here, the weight of a matching is given by Equation (Equation 2), i.e., by the sum over all weights of the matched edges.

**Definition** **5.**
*A matching M is called a *minimum-weight (maximum-weight) matching* if its weight takes a minimum (maximum) value within the set of all matchings.*

*Correspondingly, a matching is a *minimum-weight (maximum-weight) perfect matching* if its weight takes a minimum (maximum) value within the set of all perfect matchings.*


For the case of perfect matchings, one can simply map between minimum-weight matchings and maximum-weight matchings: let $W_{\max}>\max_{\left\{i,j\right\}}\left(w_{ij}\right)$ and then define ${\tilde{w}}_{ij}=W_{\max}-w_{ij}$ for all given edge weights $w_{ij}$. A maximum perfect matching for weights ${\tilde{w}}_{ij}$ can be obtained by calculating a minimum perfect matching with weights $w_{ij}$.

Since negative-weight percolation is a model of paths, next, we start with some examples of previous studies of different paths in statistical physics.

## 3. Minimum-Weight Paths

For several decades, there has been a strong interest in the statistical properties of models for paths on networks sampled from disordered graph ensembles. These paths were used to model line-like quantities, e.g., within field-theoretical studies of networks of vortex strings connect to a symmetry-breaking phase transition in [38,39,40], for non-branched polymers in random/disordered environments [41,42,43,44,45], as domain-wall excitations in magnetic alloys such as spin glasses [30,46], like vortex loops the d=3 XY model [47,48] and in high-$T_{c}$ superconductors at zero field [49,50,51,52], as  well as for the solid-on-solid model [53]. To obtain the corresponding paths, one can often formulate the problem as a combinatorial optimisation problem. Thus, exact optimisation algorithms originating from theoretical computer science are often-used tools [29,54,55,56].

To study the statistical-mechanics properties of these paths on graphs, observables and analysis concepts are often used which originate from percolation theory [57,58] and other models which involve string-like objects [59,60,61]. Concerning percolation, the basic issue is whether nodes are connected or not. The most fundamental example is 2D (random bond) percolation. For a given underlying lattice, one “occupies” a randomly chosen fraction of the edges. Sites which are connected by occupied edges are considered as clusters. Their geometric properties are analysed. As a function of the fraction of occupied edges, the observed properties will vary. This is visible in particular by a transition from a phase with rather small disconnected clusters to another phase, which is characterised by one huge cluster which extends over a finite fraction of the sites. Typically, this transition is of second order.

## 4. Negative-Weight Percolation

Now, we will actually introduce the model for *negative-weight percolation* (NWP) [62,63,64,65,66,67,68]. This model exhibits some differences in comparison with other percolation models involving string-like objects. We consider a regular lattice graph having boundary conditions (BCs) which are fully periodic. Undirected edges connect adjacent sites. To the edges, additional weights are assigned. They represent quenched random variables. We are interested in weight distributions where the drawn values can be positive as well as negative. To control the disorder, we assume that the distribution exhibits some disorder parameter ρ which we can tune to drive the system to the phase transition—for details, see Section 4.1. For any disorder realisation, the task is to compute for the lattice a configuration of closed paths (loops), with the requirement that the sum of the weights from the edges which are part of the loops attains a *global* minimum. This means in particular that there will be no loops exhibiting a positive weight. Furthermore, one additional path might be generated in the lattice, for which it is possible to specify its start and end points. For this problem, we can involve exact algorithms from algorithmic graph theory. This is beneficial compared to the application of Monte Carlo methods which are often used to study disordered systems. Monte Carlo algorithms have typically the disadvantage that they have to be run “long enough” to equilibrate the simulation. Below, we will see that the models we study here allow us in particular to apply algorithms which run in polynomial time. Thus, we are able to treat rather large instances, allowing to enhance the reliability of statistical estimates and finite-size-scaling analyses. The loops studied here exhibit the additional constraint that they are not allowed to intersect. As a result, for the NWP model, there is no natural notion of clusters. Note that the loops can be studied for arbitrary graphs, e.g., lattices of any dimensions. In any case, independent of the spatial dimension of the lattice, the studied objects are inherently line-like. Still, the loops may fill the lattice and in such a way form fractals which exhibit fractal dimensions $d_{f}$ larger than one. Because a loop is not allowed to intersect itself nor other loops on the lattice, it possesses an “excluded volume” similar to self-avoiding walks (SAWs) [58]. The problem of finding these loops can be reformulated as a NWP (also termed minimum-weight path, MWP) problem, as described in Section 4.1.

Before we go into the details, we outline here the main characteristics of NWP: most importantly, the NWP model exhibits a disorder driven phase transition of the geometrical properties of the loops. In particular, the typical loop size [62] changes suddenly at a critical threshold value of ρ—for details; see Section 5. One finds two distinct phases, depending on the value of the disorder parameter and on the lattice structure:a phase with “small” lattice loops. This means that the lengths of the loops are short as compared to the size of the system; see Figure 10a (therein, the loop “size” is taken as the projection on the lattice axes). This resembles a diluted gas of loops.another phase exhibiting “large” loops resulting in dense configurations of packed loops. The entire lattice is eventually spanned, i.e., the loops “percolate”; see Figure 10c.

In the thermodynamical limit, there exists a particular value of the parameter ρ, denoted as $\rho_{c}$, where there is a transition between these two phases; see Figure 10b.

In the following section, the simple case of a two-dimensional square lattice is considered. This allows us to state the precise definition of the NWP model. In addition, we will present the algorithm yielding path and loop configurations of minimum weight.

### 4.1. Problem Definition of the NWP Problem and Algorithms

Here, we study lattice graphs G=(V,E), *V* being the sets of nodes and *E* the sets of edges, respectively. Except in Section 5.2, where *r*-regular random graphs are considered, we study hyper-cubic lattices, where *L* is the lateral extension, exhibiting periodic boundary conditions (BCs) in all directions. We consider dimensions ranging from d=2 to 6.

N=|V| denotes the number of sites (for hyper-cubic lattices, this results in $N\;=\;L^{d}$) and m=|E| the number of (undirected) edges {i,j}∈E which connect neighbouring sites i,j∈V. For each edge {i,j}∈E, we assign a weight $\omega_{ij}$. These weights are quenched random variables which are independently identically distributed (iid). They comprise the disorder attributed to the lattice.

Two distributions that have been studied commonly are:*bimodal*±J edge-weight distributions (usually J=1) [62]
(3)P(ω)=ρδ(ω+J)+(1−ρ)δ(ω−J).The fraction of negative weights is denoted by ρ.Weights can either exhibit a unit weight (with probability 1 − ρ), or, with probability ρ, they are drawn according to a Gaussian distribution (zero mean and variance one). This is described by the following probability density:
(4)$$P\left(\omega \right)=\rho \exp\left(-\omega^{2}/2\right)/\sqrt{2\pi }+\left(1-\rho \right)\delta \left(\omega -1\right)\;.$$This distribution is called *mixed Gaussian* here.

Both types of weight distributions allow for loops L⊂E which exhibit a weight $\omega_{L}\;=\;\sum_{\{i,j\}\in L}\omega_{ij}$ being negative. The resulting configurations will be as follows: Any value of the *disorder parameter*
ρ>0 will result in, for sufficiently large lattice size, the creation of at least “small” loops exhibiting a negative weight; see Figure 10a. For sufficiently large values of the disorder parameter, there will be negative-weighted loops which span the system; see Figure 10b,c. The NWP problem is defined as follows: Let be *G* a graph attached with a set of (randomly-assigned) weights. Then, determine a loop set C with the requirement that the energy, which is given by sum $E\;=\;\sum_{L\in C}\omega_{L}$ of the weights of all loops, is minimised. As an additional constraint for the optimisation problem, any intersection of the loops is forbidden. Note that, in general, the weights of all the loops are negative. This means that the set C may also be empty (this is trivially true for the case of no negative edges, ρ=0).

The function to be optimised is the configuration energy E. The result of the optimisation process is a set of loops C. This set can be calculated within several steps, starting by constructing a suitably defined auxiliary graph based on the original graph [69]. This auxiliary graph is constructed such that a relation between minimum-weight paths (and loops) on *G* and minimum-weight perfect matchings (MWPM) on the auxiliary graph holds [16,70,71].
An auxiliary graph $G_{A}$ is created in two steps. the edges are treated first and then the nodes of the original graphs: Each edge, which joins adjacent sites within the original graph *G*, is replaced by a path made by three edges. For this purpose, one has to introduce two “additional” sites for each edge of the original graph. One of the two new edges which connects an original site to an additional site gets assigned a weight with the same value as the corresponding edge in the original graph. The other two edges of such a three-edge path get assigned a weight of zero.Next, each original site i∈V is “duplicated”. This means *i* is replaced by two nodes $i_{1},i_{2}$. Correspondingly, all their incident edges and their weights are duplicated as well. Furthermore, one additional edge $\left\{i_{1},i_{2}\right\}$ with zero weight is created to link the duplicated sites $i_{1}$ and $i_{2}$, for each pair $i_{1},i_{2}$. These edges have weight zero. The result for the sample auxiliary graph $G_{A}=\left(V_{A},E_{A}\right)$ is displayed in Figure 11II. Here, diamonds are used to depict the additional sites and circles to depict the duplicated sites. The weights assigned in the transformed graph $G_{A}$ are also included in Figure 11II. A more extensive description of the mapping can be found in the literature [30,69].Next, for the auxiliary graph, an exact MWPM is obtained by applying combinatorial-optimisation algorithms [70,72]. This matching is given by a subset $M\subset E_{A}$ with minimum weight, such that each node from the node set $V_{A}$ is incident to exactly one edge of *M*. An illustration is shown in Figure 11III. Here, solid edges display *M* for the chosen assignment of weights. The dashed edges are not part of the matching.Finally, one basically inverts the transformation (1), which allows one to obtain a configuration of negative-weighted loops C on *G* from the matched edges *M* of the auxiliary graph. To understand this, note that each matching edge which is incident to a duplicated site (circle) and an additional site (square) will contribute to a loop; see Figure 11IV. More precisely, for each loop segment on *G*, there are always two such edges in *M*. Every matching edge which connects sites of the same type (i.e.,  additional with additional or duplicated with duplicated) carries a weight of zero and is not contained in a loop concerning *G*.After the set of loop edges is constructed, one can use a depth-first search [16,69] to actually find the set C of loops. For analysis, the geometric properties of all loops can be calculated. In the example shown in Figure 11I, there exists only one loop with negative weight with $\omega_{L}\;=\;-2$. It has the length ℓ=8.

Here, just a concise description of the steps is presented that yields a minimum-weight set of loops for any lattice realisation of the disorder. Figure 11 displays the three fundamental steps, as detailed next:

The fact that via the calculation of a MWPM minimum-weight loops are obtained can be understood in the following way: For a perfect matching, all nodes have to be matched. This is in particular true for the additional (diamond) nodes. Thus, there are two cases:An additional node in $G_{A}$ is matched with a duplicated node (see Figure 12I). In this case, the other duplicated node must be matched to an additional node as well because his “twin” is not available any more. This exactly models the path property: a path which enters a node (modelled by a pair of duplicated nodes in $G_{A}$), corresponding to an additional-duplicated match, must leave the node again, i.e., there must be a second additional-duplicated match. For this reason, the weights are attached to the edges which connect additional to duplicated nodes.An additional node from $G_{A}$ is matched to a second additional node. This represents the case that a path segment is absent here. This is why we assign the weight 0 for edges connecting additional to additional nodes. Such a match does not influence how the duplicated nodes are matched (see Figure 12II). It is possible that a node of *G* does not participate in a path or loop, in which case all adjacent additional nodes in $G_{A}$ are matched to additional nodes (see Figure 12III). In this case, to obtain a perfect matching, the two duplicated nodes must be matched to each other.

Note that the result of the calculation is a collection C of loops such that the total loop weight, and consequently the configuration energy E, is minimised. Hence, one obtains a global collective optimum of the system. Obviously, all loops that contribute to C possess a negative weight.

The full calculation of C is performed in a running time which increases only polynomially in the system size. Trivially, the construction of the auxiliary graph is done in linear time O(N+M), where N=|V| is the number of nodes in the original graph and M=|M| the number of edges in the original graph. The auxiliary graph exhibits 2N+2M nodes and N+5M edges. For the calculation of the MWPM, fast polynomial running algorithms exits. The *Blossom* algorithm by Edmonds [73] has a worst-case running time of $O(N^{2}M)$. This results in a total running time for the calculation of the optimum set C of loops asymptotically bounded by the polynomial running time of the Blossom algorithm. By using a fast implementation [72], large systems can be solved in practice, e.g., square lattices exhibiting 512×512, i.e., $O(10^{5})$, nodes.

In addition, note that, from a procedural point of view, the problem of finding a ground-state spin configuration for a planar 2D triangular random-bond Ising model (i.e., including the spin glass introduced in Section 1) is directly equivalent to the NWP problem on a suitably weighted hexagonal graph [74], thus highlighting the close relation of both problems.

### 4.2. Paths

Above, we have explained how a set of loops is obtained: at each node of *G*, there is either no path, or there is a path going in and a path going out. This is exactly the property of a set of loops. In case one wants to induce additionally a minimum-weight path (MWP) which runs between two selected nodes *s* and *t*, one has to create a slightly modified auxiliary graph. For this purpose, the nodes *s* and *t* will not be duplicated, but still all other nodes, as shown in Figure 13I. According to step (2) above, the MWPM will be calculated for the auxiliary graph in the same way. According to step (3), it will result in a path of minimum weight between *s* and *t*. In addition, a possibly empty set of loops is obtained. An example is shown in Figure 13II,III. Here, the same assignments of weights like in Figure 11I are used, but now for the calculation of a path of minimum weight.

In Figure 14, sample results, which consist of a path and a set of loops, are displayed. As obvious, when increasing the value ρ of the parameter controlling the disorder, the path and the loops get longer. In addition, here the percolation phenomenon when increasing the value of ρ is visible.

Note that, for finding minimum-weight paths, it is not possible to somehow change the weights such that all of them are positive, and use a standard shortest path algorithm to obtain the (same) minimum-weight paths. This is shown by the example in Figure 15, where the minimum-weight (“shortest”) path changes, after all edge weights were shifted up by the amount of the minimum weight. Thus, for paths including negative weights, in particular when negative-weight cycles might appear, one has to use the more complicated mapping to an auxiliary graph, followed by the calculation of a MWPM.

### 4.3. Other Graphs

The above presentation is for two-dimensional lattices. NWP was defined actually when considering the calculation of domain-walls for planar Ising spin glasses, as explained in Section 1. Nevertheless, the NWP problem can be defined for arbitrary graphs, and, in particular, higher dimensional lattices. The transformation procedure yielding the auxiliary graph remains exactly the same, since it is defined for arbitrary graphs anyway. In addition, the calculation of the matchings is not altered, thus a polynomial-time problem remaining. Furthermore, the algorithmic procedure naturally extends to *r*-regular random graphs, discussed in Section 5.2 below.

Note only that, for higher dimensions or arbitrary graphs, the calculation of minimum-weight paths does *not* allow for the calculation of spin-glass ground states. The latter is, except for the planar 2D case, an NP-hard (nondeterministic polynomially-hard) problem [75].

### 4.4. Intersecting Loops or Paths

The construction of the auxiliary graph $G_{A}$ presented in Section 4.1 allows at each node of *G* at most one path entering and leaving. Technically, the reason is that the nodes of *G* are duplicated, thus, they can be either matched to each other (no path) or both to “additional” nodes (one path entering and leaving). One can allow for multiple paths, e.g., two, by replacing each node of the original graph by four nodes. Now, all edges are replaced by four edges and, in addition, each quadruple of four nodes in $G_{A}$ is connected among each other (leading to a tetrahedron). Now, for the nodes of the tetrahedron, there are only three cases: (I) they are matched among each other, corresponding to no path. (II) Two of them are matched to each other and the other two are matched to additional nodes. This corresponds to one path entering and leaving the original node. (III) All four tetrahedron nodes are matched to additional nodes. This correspond to two paths entering and leaving the original node. Generalisations to higher dimensions are straightforward.

### 4.5. Directed NWP

NWP can also be defined on directed graphs. For this case, the definition of the auxiliary graph has to be slightly modified; see Figure 16. What remains the same is that each edge in the original graph *G* is replaced by two additional nodes and three edges in $G_{A}$. In addition, each node in *G* gets duplicated. The only difference is that the duplicated nodes are distinguished (“white” and “black”) and that, for the heads of each edge in *G*, there is in $G_{A}$ only a connection between an additional node and a white node, while, for the tails, there is only a connection between an additional node and a black node. This guarantees that the MWPM corresponds to either no path through an edge, or to a path where exactly one incoming and one outgoing edge is used.

## 5. Results for NWP

In this section, we present the main results obtained over the past decade for NWP. First, a very brief summary is given: we have studied NWP on hyper-cubic lattice graphs for dimensions ranging from two to seven [62,64]. We have analysed the occurring phase transitions using finite-size scaling (FSS) and obtained in particular the critical exponents which are characteristic of the transitions. Here, we summarise the corresponding studies. For two dimensions, the exponents turned out to be universal with respect to different lattice geometries and disorder distributions. Furthermore, the values for the exponents were clearly different from the case of standard bond or site percolation. The studies furthermore indicate that the upper critical dimension of NWP is $d_{u}=6$ (see Section 5.1). This result was confirmed by a study of NWP regular random graphs, which describes the mean-field case of the NWP model. Within an analytic approach, the relationship of polymers in random media to NWP was utilised; see Section 5.2. The consequences of dilution for the NWP phase transition were studied for the case of two dimensions (shown in Section 5.3). It was also investigated whether two-dimensional NWP can be modelled by Schramm–Loewner evolution, as presented in Section 5.4). Most recently, NWP was analysed for the case of directed graphs, as presented in Section 4.5. In the limit of high disorder, fully and densely packed loop configurations were investigated [76]. Additionally, the algorithm to obtain NWP configurations was modified such that it allows us to obtain ground states for the random bond model on planar triangular lattices, i.e., for spin glasses [74]. Furthermore, non-integer dimensions like d=2.32 can be addressed with the Migdal–Kadanoff approximation approach, where NWP was studied on hierarchical disordered lattices [65]. Finally, several variants of local optimisation algorithms involving dynamical random-walk approaches were applied to the two-dimensional NWP problem [68]. However, to keep the text compact, these latter studies are not reviewed further here.

### 5.1. Percolation Transitions

By applying numerical simulations, we characterised the percolation transition occurring for NWP. We analysed the geometry of the minimum-weight loops and paths and by measuring observables borrowed from standard percolation [57,58]. The NWP loop setup was considered on hyper-cubic lattice graphs in dimensions d=2 through 7 [64]. As an example, in Figure 17, the spanning probability $P_{L}^{s}$ for loops on two-dimensional lattices with mixed Gaussian disorder according to Equation (Equation 4) is shown, for different system sizes in the range L∈[24,128]. Here, a system is considered spanning if it exhibits at least one loop whose projection on an axis (*x*-axis or *y*-axis in two dimensions) covers the full axis. As the inset shows, the spanning probabilities increase with increasing fraction ρ, and the curves for different system sizes roughly intersect at one point near ρ=0.34, which signifies the percolation transition point. In order to determine the critical point more precisely and to estimate the value of critical exponent ν describing the divergence of the correlation length, an FSS analysis [58,77,78] can be performed. Here, it is assumed, as is common for second order phase transitions, that the percolation or spanning probability follows a scaling behaviour according to
(5)$$P_{L}^{s}∼f\left[\left(\rho -\rho_{c}\right)L^{1/\nu }\right].$$

Thus, when rescaling the ρ-axis according to $\left(\rho -\rho_{c}\right)L^{1/\nu }$, with correctly chosen values for $\rho_{c}$ and ν, the data for different system sizes should follow a universal curve. This is called a *data collapse*. Thus, if the correct values for $\rho_{c}$ and ν are not known, one can estimate them by starting with some guesses and iteratively adjusting them such that the data points *collapse* onto a universal curve as best as possible. This can even be achieved automatically, e.g., by using the autoScale.py Python tool [79]. In this case, the data collapse yielded the best quality for values ρ=0.340(1) and ν=1.49(7). The quality of the collapse, i.e., the mean deviation of the data points of the rescaled curves, normalised by the error bars, was S=0.91 which is very good.

In addition, other quantities can be measured and analysed by finite-size scaling. Two important quantities are based on the length *ℓ* of the largest loop in the system. Thus, the probability $P_{L}^{\infty }\equiv \langle \ell \rangle /L^{d}$ that any edge belongs to a percolating loop and the susceptibility $\chi \equiv L^{-d}\left(\langle \ell^{2}\rangle -{\langle \ell \rangle }^{2}\right)$ were studied. Their behaviour near the critical point is governed by the critical exponents β and γ, respectively. They can be obtained by applying analogous FSS assumptions
$$\begin{array}{ccc} P_{L}^{\infty } & ∼ & L^{-\beta /\nu }g\left[\left(\rho -\rho_{c}\right)L^{1/\nu }\right]\\ \chi & ∼ & L^{\gamma /\nu }h\left[\left(\rho -\rho_{c}\right)L^{1/\nu }\right]\;. \end{array}$$

An analysis of the data via data collapse (not shown here) yielded β=1.07(6) and γ=0.77(7). Furthermore, the fractal dimension of the percolating loops can be studied, i.e., how their average length depends on the system size via $\langle \ell \rangle ∼L^{d_{f}}$, yielding (not shown here) $d_{f}=1.266\left(2\right)$ in this case. The so far-determined values are compatible with the standard percolation-theory hyper-scaling relations γ+2β=dν (d=2 here) and $d_{f}=d-\beta /\nu$.

In addition to a possible large percolating loop, there are always many small and medium sized non-percolating loops, like in ordinary percolation, where there are always small clusters. One can analyse, e.g., the distribution of areas enclosed by the loops [76]. Here, we focus on the distribution of the lengths of the loops. The result for two-dimensional NWP with mixed Gaussian disorder is shown in Figure 18. By the analysis of the distribution of loop sizes [66], two more critical exponents come into play. Right at the critical point $\rho =\rho_{c}$, the distribution exhibits a power law, governed by the exponent τ. Away from the critical point, large loops are exponentially suppressed. This can be modelled by a *loop tension*
$T_{L}$. All together, this leads to the following scaling form for the distribution of loop sizes *ℓ*:(6)$$n_{\ell }\left(\rho \right)\;∼\;\ell^{-\tau }\exp\left(-T_{L}\left(\rho \right)\ell \right)\;.$$

The loop tension vanishes at the critical point to allow for the pure power-law behaviour. The behaviour of the loop tension as a function of the critical point is expected, like in ordinary percolation [58], to follow a power law as well
(7)$$T_{L}\;∼\;{\left|\rho -\rho_{c}\right|}^{1/\sigma }\;$$
described by the “loop-size cut-off” exponent σ. For the case shown above, a fit to the power-law distribution at $\rho \approx \rho_{c}$ yielded τ=2.59(3). Using this value, by fitting Equation (Equation 6) for various values of ρ resulted in $T_{L}$ as a function of $|\rho -\rho_{c}|$, as shown in the inset of Figure 18. Here, a high-quality power-law is also visible, resulting in σ=1.89(7). These two values are consistent with scaling relations $\tau \;=\;1\;\;+\;\;d/d_{f}$ and $\nu d_{f}=1/\sigma$.

Note that similar simulations were also performed for different types of lattices (e.g., having hexagonal structure) and bimodal disorder Equation (Equation 3). The results for the critical exponents turned out to be compatible within error bars with the results presented so far. Furthermore, a similar study was performed for the NWP minimum-weight path setup in 2D only [62]. Here, a path was forced into the system by the mechanism explained in Section 4.2 to run from the bottom to the top boundary. The path was considered to be percolating if its projection on *x*-axis covered the full *x*-axis (it covers the *y*-axis by construction). The results for critical point and critical exponents were obtained in the same way and turned out to be again compatible with the corresponding results for loops. Thus, it can be concluded that the behaviour with respect to lattice types, disorder types and with respect to loop or path negative weight percolation is universal [62].

For standard percolation [58], conformal field theory yields the exact value ν=4/3≈1.33, which is slightly incompatible with the above stated result. For other exponents, the situation is much clearer because the standard percolation values for β=5/36≈0.14 and γ=43/18≈2.39 are very different from the values found for NWP. Thus, NWP is apparently in a different universality class than percolation. Note that also other well-known universality classes are clearly different, e.g., the ferromagnetic Ising exhibits ν=1 and the *q*-state Potts model ν=5/6 (q=3) and ν=2/3 (q=4). On the other hand, the critical exponents are very similar to the values obtained for the 2D random-bond Ising model [25]; for a discussion, see Ref. [62].

Similar numerical studies have also been performed in higher dimensions [64] d=3−7 and have been analysed in the same way as reported for the 2D case reported above. The resulting critical exponents are summarised in Table 1.

The main result was that, for all dimensions studied, the critical exponents are clearly different from the values which are found for commonly studied percolation models of strings. They again are compatible with the hyper-scaling relations γ+2β=dν and $d_{f}=d-\beta /\nu$, which were mentioned already above.

On the other hand, for three dimensions, the values for the critical exponents of the strongly screened vortex glass model [51] are very similar to the exponents found for NWP. In addition, all numerical results obtained in dimensions larger than three indicate that the upper critical dimension for NWP is $d_{u}=6$. This motivated a study on mean-field graphs, which is summarised below in Section 5.2.

In addition, the MWP algorithm was empirically investigated with respect to the computational effort [62]. The running time on average just increases when increasing the disorder strength. Interestingly, the fluctuations of the running time show a pronounced peak close to the percolation transition $\rho_{c}$, and the height of this peak increases when increasing the system size *L*. Thus, the physical phase transition is also visible in the algorithmic behaviour.

### 5.2. Mean Field Behaviour of the NWP Model

The behaviour of NWP above the upper critical dimension was further elaborated within a different study [80]. For this purpose, r=3-regular random graphs were investigated, i.e., where there are exactly three neighbours per node without any other regularity. Again, the behaviour of the resulting minimum-weight paths and loops on these graphs was analysed. Because there is no spatial structure, hence the graphs are infinite-dimensional $d\geq d_{u}$, one obtains the mean-field exponents directly. The disorder distribution was bimodal as described by Equation (Equation 3).

On the other hand, the absence of spatial structure does not allow for the definition of a criterion for spanning paths or loops. Thus, in order to detect the percolation threshold, among others the following approach was used: first, the *diameter*
$R_{N}$ was calculated for every graph. The diameter is the longest among all shortest paths, where the maximisation is over all pairs of nodes and the minimisation is over all paths between two given nodes. Note that path lengths are measured in terms of the number of edges.

Next, for a pair of nodes which have distance $R_{N}$, the MWP between the two nodes was calculated using the matching approach. For two limiting cases, the behaviour of averaged length 〈ℓ〉 of the minimum-weight path is easy to guess. For ρ→1, one should see 〈ℓ〉∝N because the path aims at picking up as many negative weights as possible. For ρ→0, where no negative weights are available, an $\langle \ell \rangle \propto R_{N}$ is to be expected. These two regimes should be separated by a threshold $\rho_{c}$. Right at the threshold, the loops should behave like standard random walks. Thus, at $\rho_{c},$ the minimum-weight paths should scale as $\langle \ell \rangle \propto R_{N}^{2}$. From this scaling, the transition could be estimated to be in the range [0.04,0.25].

Next, the critical point has precisely been determined [80] by calculating the probability $P_{N}$ that the weight $\omega_{p}$ of a path is below zero and performing a finite-size scaling analysis. As the inset of Figure 19 shows, the probabilities $P_{N}$ do not intersect at one single point, but a small systematic shift was observed. This was taken into account, by writing the finite-size critical point, as in standard FSS for second order transitions, as $\rho_{1}\left(N\right)=\rho^{c}+aN^{-\phi_{1}}$. This form was used for the finite-size scaling in $P_{N}=f\left[\left(\rho -\rho_{q}\left(N\right)\right)N^{-1/\nu }\right]$ and all parameters $\rho^{c}$, *a*, $\phi_{1}$ and ν were varied to yield an optimum data collapse. As a result, also from the analysis of other measurable quantities (not shown here), $\rho_{c}=0.075\left(1\right)$ and ν=3.0(1) was found. Note that this value for ν matches immediately the values stated for d=6 in the above table, since the $L^{1/\nu }$ scaling behaviour corresponds to $N^{1/d\nu }$ and at d=6 one has (see table) dν=6×0.5=3. In addition, at $\rho_{c}$, precisely the $\langle \ell \rangle \propto R_{N}^{2}$ behaviour could be observed.

Furthermore, several other critical exponents have been found. In particular, β=1.82(1) being roughly compatible with the value β=1.92(6) for d=6 was obtained. Note that these error bars denote only the statistical uncertainty and that strong finite-size effects were observed in the simulations, such that also a larger value for β could not be ruled out [80]. For all cases, the obtained values were in agreement with the exponents which were previously found for the six-dimensional hypercube. Furthermore, the position of the critical point agreed very well with results obtained by analytic means, exploiting the similarity between polymers in random media and the NWP problem. In particular, the calculation was performed within the replica symmetric cavity approach. This further allowed to analytically determine a critical exponent, β=2, which is also roughly compatible with the numerical results, given the uncertainty mentioned above. Note that it could be interesting to study as a generalization finite-dimensional lattice with added infinite-range bonds. This might lead to continuously varying exponents, interpolating between finite-dimensional and mean-field behavior, as it was found for standard percolation [81].

### 5.3. Phase Transitions in Diluted Negative Weight Percolation Models

For the special case of two-dimensional lattices, the effect of dilution was studied. Thus, a random fraction of the edges was absent in two ways for each realisation, changing the (effective) topology of the lattices. Again, standard observables from percolation theory [57,58] were employed in connection with FSS analysis. In this way, the critical boundaries in the dilution-disorder phase diagram could be identified. This was done for two variants of the dilution:For type I dilution, the disorder is realised by selecting a fraction $p_{I}$ of edges where the weights $\omega_{ij}$ are zero. Thus, the paths or loops can use these edges without changing the weight.For type II dilution, a random fraction $p_{\mathrm{II}}$ of edges is actually *absent*. Thus, no paths or loops can run there.

The results revealed that, in contrast to type I, type II dilution modifies the value of the critical exponents of NWP. By performing an enhanced analysis for one selected critical point, i.e., $\left(\rho ,p_{\mathrm{II}}\right)\;=\;\left(1.0,0.4998\right)$, it was furthermore confirmed that the well known scaling relations are fulfilled. Additionally, the fractal dimension $d_{f}$ obtained for NWP at that particular critical point is compatible with the value for self-avoiding walks on a regular lattice ($d_{f}^{\mathrm{SAW}}=4/3$). This indicates that NWP at $\left(\rho ,p_{\mathrm{II}}\right)\;=\;\left(1.0,0.4998\right)$ is in the same universality class as two-dimensional self-avoiding walks.

### 5.4. 2D NWP and SLE

For the NWP variant where a minimum-weight path and loops coexist, the question of whether the paths might be described by the so-called *Schramm–Loewner evolution* (SLE) [1,82,83] was studied [67]. To understand what SLE means, next a concise summary of the basics is presented: the *chordal* variant of SLE models a self-avoiding curve $\gamma_{t}$ (“time” $t\in R^{+}$) which is located in the upper half plane H. Note that H can be mapped on to any other simply connected planar domain by a conformal transformation. The origin of the coordinate system is the starting point of $\gamma_{t}$. It grows as a function of the time *t* to infinity. For convenience, the two-dimensional plane is described as complex numbers z∈C. One considers the set of points consisting of $\gamma_{t}$ plus all points which are disconnected from infinity by $\gamma_{t}$. A conformal mapping $g_{t}\left(z\right)$ is defined such that it maps for each time *t* the complement of this set on to the full upper half plane.

Therefore, the interface between reachable and unreachable area is mapped to the real axis and $g_{t}\left(z\right)$ maps the growing tip of the curve for each time to a point $a_{t}$ on the real axis. Hence, $a_{t}$ describes a continuous movement on the real axis. One fundamental property is that the function $g_{t}\left(z\right)$ follows a certain ordinary differential equation, which is also called the Loewner equation:(8)$$\frac{dg_{t}\left(z\right)}{dt}=\frac{2}{g_{t}\left(z\right)-a_{t}}.$$

Let us consider the special case that $a_{t}$ is proportional to a one-dimensional Brownian motion $B_{t}$, i.e., $a_{t}=\sqrt{\kappa }B_{t}$. One can show that the corresponding curve $\gamma_{t}$ is conformal invariant [84]. This is the case which is called SLE. Therefore, any curve which can be modelled by SLE is characterised by a single value, the so-called *diffusion constant*
κ.

During the past 20 years, a lot of curves occurring in physical planar systems have been identified as SLE curves. Examples for SLEs are spin cluster interfaces for the Ising model considered at the criticality (κ=3) [85], cluster boundaries at the percolation threshold (κ=6), and interfaces of vortex clusters in two-dimensional turbulence (κ=6) [86]. In addition, the behaviour of two-dimensional spin-glasses seems to be compatible with SLE, as numerical studies indicate [87]. A thorough introduction and recent results about SLE (particularly for random models) can be found in a review [88].

For NWP, the situation seems to be different, i.e., not compatible with NWP being SLE. Several numerical studies which consider the geometry of the NWP paths result in different values for the diffusion constant κ.

One way to obtain an estimate for κ is the application of Schramm’s left-passage formula (SLPF). The SLPF states the probability that a SLE curve will pass to the left of an arbitrary given point z=x+iy, which is located in in the upper half plane (again addressed via complex numbers) provided that the curve connects the origin of the coordinate system to infinity. The following equation holds [89]:(9)$$\begin{array}{cc} P_{\kappa }\left(z\right)= & \frac{1}{2}+\frac{\Gamma (4/\kappa )}{\sqrt{\pi }\Gamma \left(\left(8-\kappa \right)/2\kappa \right)}\\ & \;\times {}_{2}F_{1}\left(\frac{1}{2},\frac{4}{\kappa };\frac{3}{2};-{\left(\frac{x}{y}\right)}^{2}\right)\frac{x}{y}\;. \end{array}$$

Here, ${}_{2}F_{1}$ is the Gaussian hyper-geometric function and Γ denotes the gamma function. Please note that $P_{\kappa }\left(z\right)$ is independent from the distance of the point *z* from the origin. Therefore, the dependence of the fraction Re(z) divided by Im(z) can be replaced by a dependence on an angle. In Figure 20, a comparison of the numerical results with the analytical prediction is shown, for the value of adjusted to $\kappa^{★}\;=\;3.343$ which yields the best agreement, which is actually perfect relative to the error bars [90] (see inset).

On the other hand, within SLE, the value of κ can be determined from the fractal dimension $d_{f}$ of the paths by $\kappa =8(d_{f}-1)$ for κ<8 (for κ>8, there is no unique relation, all these curves are space filling with $d_{f}=2$). For NWP, a value of $\kappa^{f}=1.283\left(2\right)$ was found, which is clearly different from the value obtained via the SLPF. As a consequence of the different values for κ, NWP cannot be described by SLE because, otherwise, the value of κ would be unique when obtained by different methods assuming SLE. As pointed out [67], the reason for this non-compatibility with SLE might be that the so-called “Markov-property” does not hold [1].

### 5.5. Directed NWP

Recently, NWP was studied on directed graphs [91]. The algorithmic procedure for directed graphs is explained in Section 4.5. In addition, here a disorder-driven percolation transition exists, very similarly to the undirected case. Nevertheless, the finite-size scaling is more involved because directed percolation problems are anisotropic. The system is governed by a main (or “time”) direction, which is given by the sum of the directed basis vectors of the lattice (here from top to bottom). The anisotropy arises because the objects, like clusters, scale differently parallel and perpendicular to the main direction. This can be seen in Figure 21, where it is shown how the correlation length parallel and perpendicular to the main direction is calculated (left). In anisotropic systems, the correlation lengths diverge with different critical exponents $\nu_{∥}$ and $\nu_{⊥}$. In order to avoid complicated shape effects in simulations, it is necessary to fix the aspect ratio $L_{∥}/L_{⊥}^{\nu_{∥}/\nu_{⊥}}$ [92]. Therefore, before running proper simulations, the ratio $\nu_{∥}/\nu_{⊥}$ has to be determined by $\xi_{⊥}\left(L_{∥},L_{⊥}\;\to \;\infty \right)∼L_{∥}^{\nu_{⊥}/\nu_{∥}}$ first (Figure 21, right) [93]. A detailed analysis [91] exhibits that two-dimensional directed NWP is in a universality class different from undirected NWP and also different from directed standard percolation.

## 6. Conclusions

In this review, we presented a summary of recent work on the NWP problem, i.e., a lattice-path model that exhibits a phase transition which is driven by disorder. The model is motivated by the calculation of ground states and domain walls for planar spin glasses. By abstracting from the spin-glass problem, NWP can be defined on arbitrary graphs, i.e., for lattices in any dimension. Most work published so far is focused on the variant of the NWP model where only loops are present. There, for any disorder realisation, one measures the configurational properties of a set of loops which exhibits a global minimum total weight. In addition, a variant exists where a path plus a set of loops is allowed. In contrast to percolation or to other models with path- or loop-like objects, the existence of negative weight leads to a different behaviour and thus to different universality classes.

The existence of negative weights also prohibits the direct application of standard algorithms, like shortest-path algorithms. Nevertheless, a mapping of the NWP model to an associated minimum-weight perfect matching problem allows one to obtain such GS configurations by means of exact algorithms in polynomial time. This enables one to study large systems whilst ensuring reliable statistics.

The behaviour of NWP depends on the value of the characteristic disorder parameter ρ: below a critical value $\rho_{c}$, i.e., for a small fraction of negative edge weights for a given lattice, only a few loops exist and a system spanning path, if enforced, will have positive weight. As ρ exceeds a first threshold value $\rho_{c}$ (the numerical value depends on the type of disorder, the lattice type, the dimension, and on a possible dilution) system spanning, i.e., percolating loops will appear and the path will typically exhibit a negative weight. Interestingly, the fluctuations of the running time of the matching algorithm will also be strongest close to the phase transition. For NWP, typical percolation quantities can be measured like percolation probability, size of the largest cluster, distribution of small loops, and fluctuations of these quantities. The model has been studied in various dimensions, also for mean-field graphs, and an upper critical dimension of $d_{u}=6$ was found, confirmed by analytical calculations using a cavity approach. For d=2, it was also shown that NWP does not belong to the class of SLE curves. Recently, NWP was also studied in a directed variant, and again a different behaviour with respect to the critical exponents has been found.

To summarise, the NWP model represents a spin glass-based non-standard optimisation and percolation problem. It exhibits a rich critical behaviour and several limiting cases that, on their own, represent intriguing models considered in the field of statistical physics of disordered systems.

For future studies, it would be promising to explore further the relationship to polymers in disordered media, which was already exploited for the analytical study. Furthermore, it would be very interesting to study the model at finite temperatures, which should be easier because the ground states are numerically readily available. Finally, the dynamic behaviour, e.g., ageing, could exhibit interesting properties, which one might be able to understand better on the background of the available ground states.

## Figures and Tables

**Figure 1 entropy-21-00193-f001:**
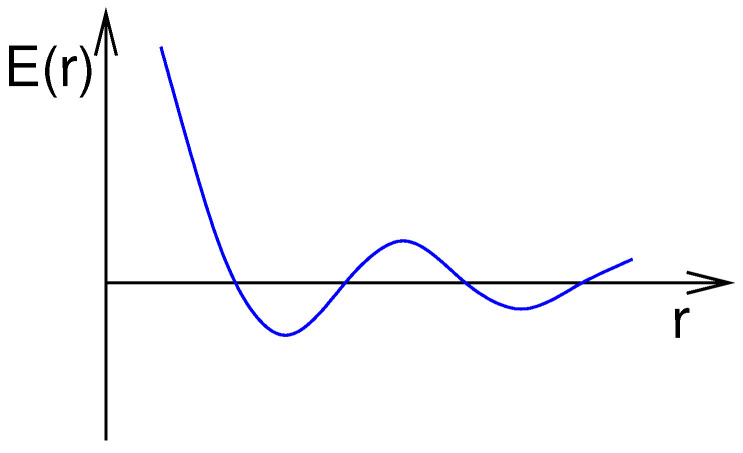
Energy of two spins placed at distance *r* coupled through a cloud of conducting electrons, yielding the RKKY (Ruderman, Kittel, Kasuya, Yosida) interaction.

**Figure 2 entropy-21-00193-f002:**
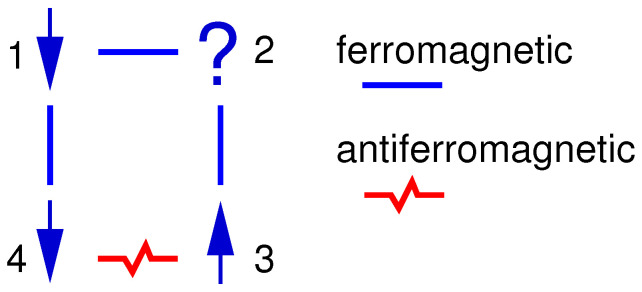
The ground state of a frustrated system consisting of four spins. Ferromagnetic interactions (“bonds”) are represented by straight lines. Thus, the interactions favour parallel orientation of the spins. An antiferromagnetic interaction is shown as a zigzag line. Independent of the orientation spin 2, one of its incident bonds is not satisfied. On the other hand, bonds 3-4 and 1-4 are satisfied.

**Figure 3 entropy-21-00193-f003:**
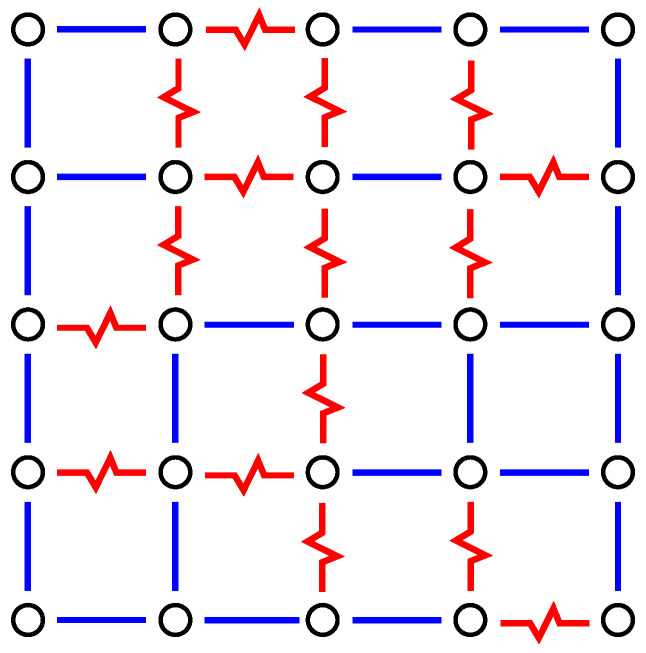
Example for a realisation of a two-dimensional spin glass (free boundary conditions). The Spins are located on the sites of a square lattice. Nearest neighbour spins interact either through a ferromagnetic (straight blue line) or antiferromagnetic (jagged red line) bond. For this example, a bimodal bond distribution is assumed, which means all bonds exhibit $|J_{ij}|=J$.

**Figure 4 entropy-21-00193-f004:**
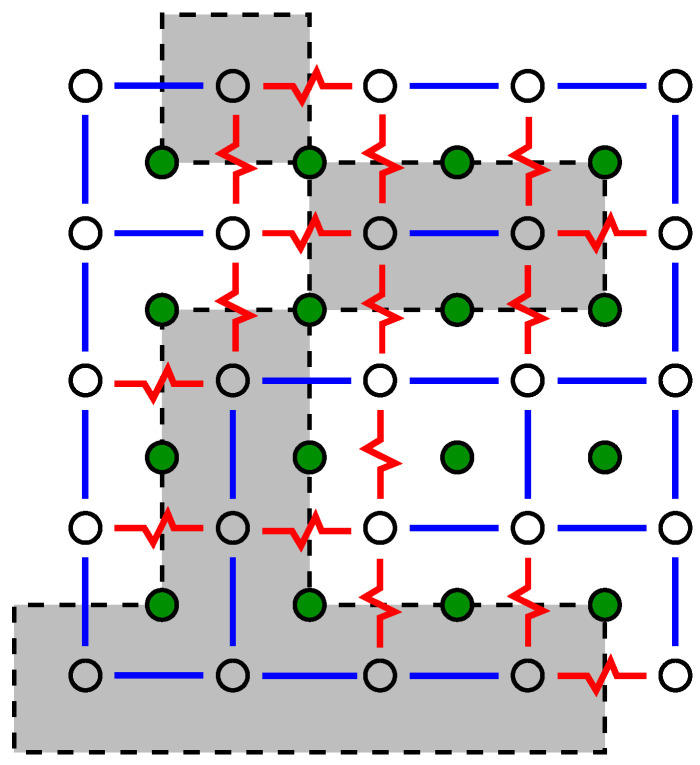
The same two-dimensional realisation as shown in Figure 3. Spins are located on the sites denoted by open circles. Filled circle symbols denote the sites of the dual lattice. Cycles on the dual lattice correspond to closed domain walls in the original lattice. Those cycles which exhibit a positive energy are indicated by dashed lines plus a grey colour for the enclosed areas.

**Figure 5 entropy-21-00193-f005:**
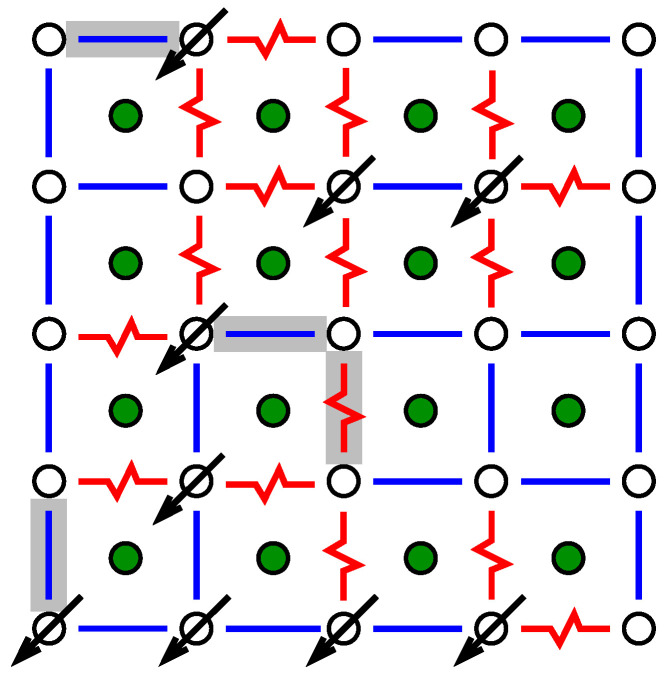
The same two-dimensional realisation as shown in Figure 3. Spins are located on the sites denoted by open circles. Filled circles denote the sites of the dual lattice. A ground state is depicted, where spins pointing downwards are shown, all other spins point upwards. The bonds which are not satisfied are marked grey. There does not exist a closed domain wall with positive energy.

**Figure 6 entropy-21-00193-f006:**
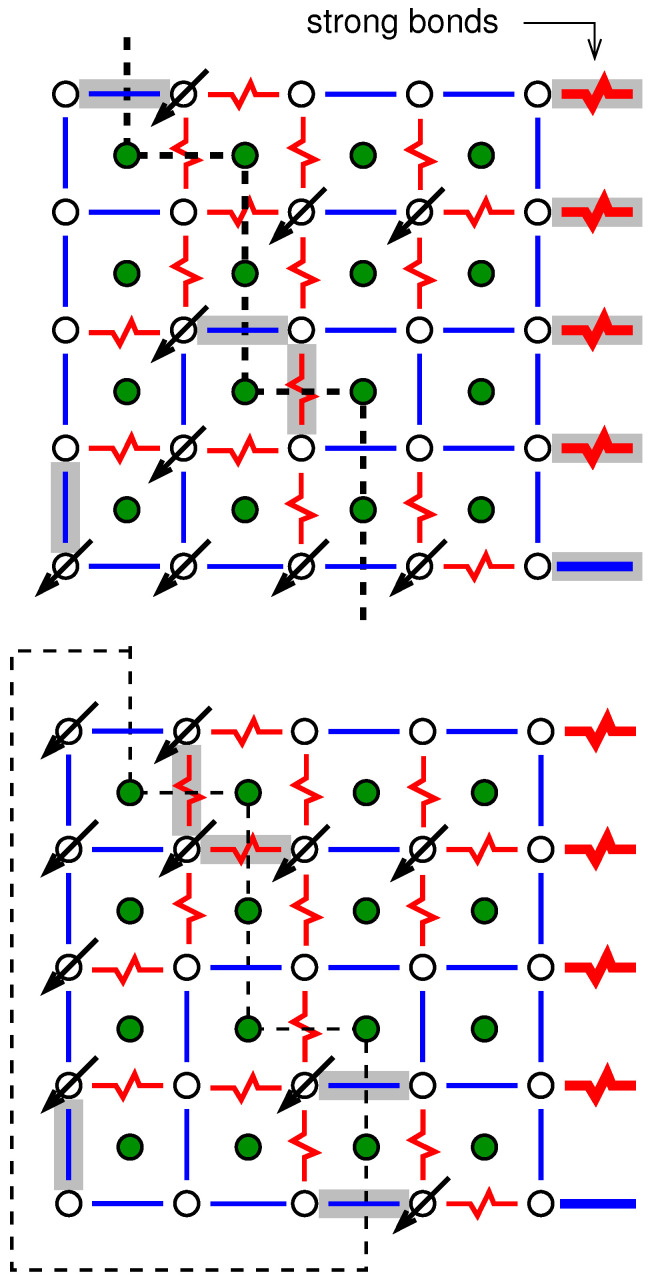
**Top**: An additional column of very *strong* bonds (shown at the right with very thick lines) is added for the example spin glass as shown in Figure 3. The bonds are *not* compatible with the GS configuration. This will force the spins in the first and last column to flip relative to each other and force a domain wall of minimum energy (dashed line) into the system. **Bottom**: New ground state for the modified system. The spins on the left inside the marked area have been flipped. In both cases, again, spins are located on the sites denoted by open circles. Filled circles denote the sites of the dual lattice. The bonds which are not satisfied are marked grey.

**Figure 7 entropy-21-00193-f007:**
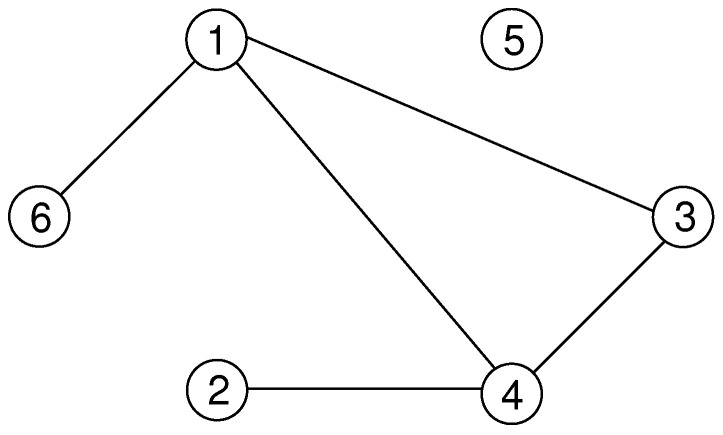
An undirected graph.

**Figure 8 entropy-21-00193-f008:**
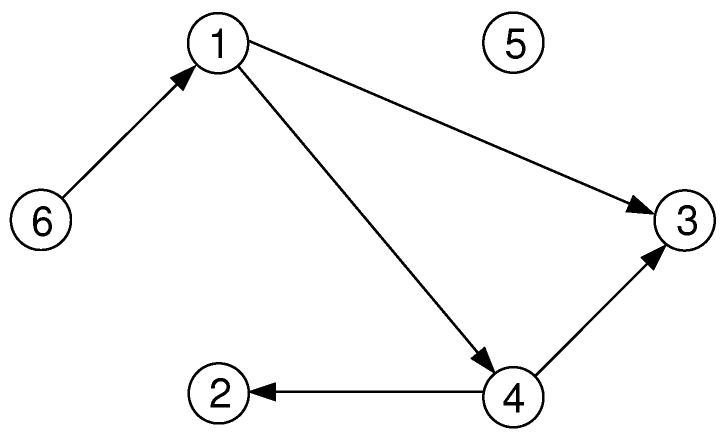
A directed graph.

**Figure 9 entropy-21-00193-f009:**
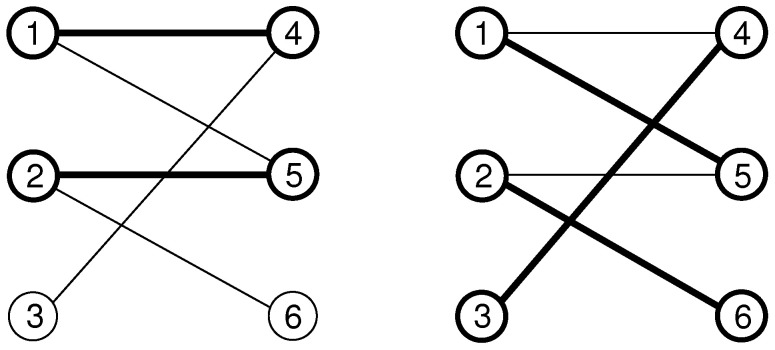
Illustration of matching for a sample graph of six nodes and four edges. In the left, the matching consist of two edges {1,4} and {2,5}. Matched edges and matched nodes are shown in bold lines. On the right, a perfect matching is shown, i.e., all nodes are matched.

**Figure 10 entropy-21-00193-f010:**
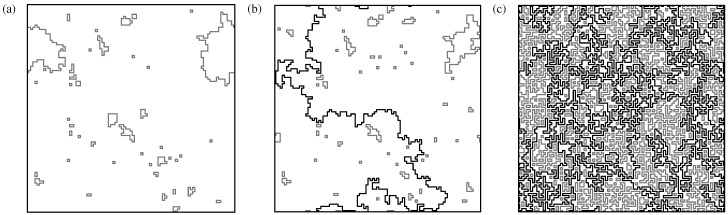
Illustration of the NWP phase transition. Sample configurations of loops on a square grid for L=96 side length, with periodic boundary conditions. (Non) percolating loops are shown in black (grey). The configurations are taken for different values of the parameter ρ which controls the disorder. (**a**) $\rho <\rho_{c}$, (**b**) $\rho \approx \rho_{c}$, and, (**c**) $\rho >\rho_{c}$. Beyond the critical point $\rho_{c}$, in the thermodynamic limit L→∞, there exist paths which span the lattice in any direction which exhibits periodic boundaries.

**Figure 11 entropy-21-00193-f011:**
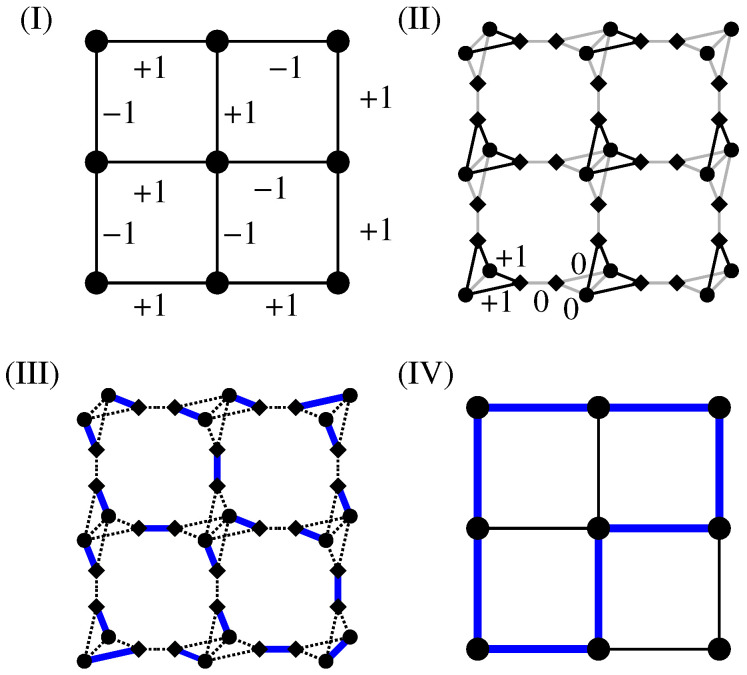
Examples of the main steps of the algorithmic procedure: (**I**) the original lattice *G* together with weights of edges; (**II**) the auxiliary graph $G_{A}$ with corresponding assignment of weights. Black edges exhibit the same value of the weight as the corresponding edge in the original graph and grey edges exhibit a weight of zero. Diamond symbols are used to mark the “additional” sites; (**III**) minimum-weight perfect matching (MWPM) *M*: the matched edges are shown in bold and unmatched edges using dashed lines, and (**IV**) loop configuration (bold edges) that is the result of the MWPM shown in (**III**).

**Figure 12 entropy-21-00193-f012:**
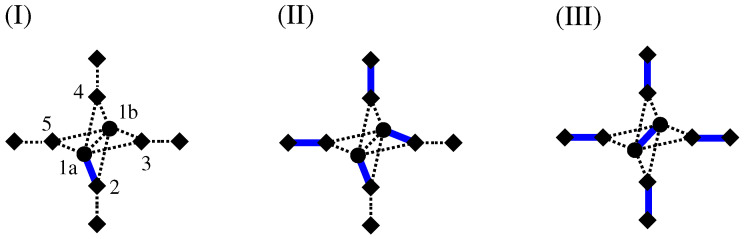
How the matching models detects loops: (**I**) the additional node 2 (shown as diamond) is matched to a duplicated node 1a (shown as circle). This means the duplicated node 1b must be matched to an additional node (either 3, 4 or 5) as well. (**II**) When 1b is matched, e.g., with node 3, the other additional nodes connected to 1a or 1b must be matched with additional nodes as well because 1a and 1b are matched already. (**III**) It is possible that around nodes 1a,1b all additional nodes are matched with additional nodes. In this case, 1a and 1b are matched with each other.

**Figure 13 entropy-21-00193-f013:**
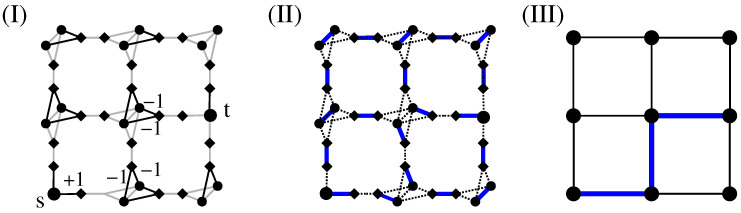
Illustration of the algorithmic procedure for s-t paths: (**I**) auxiliary graph $G_{A}$. A modified mapping is used to create a minimum-weight path which connects the two nodes *s* to *t*. These two nodes are not duplicated. Black edges obtain the same weight values (partially shown as numbers here) as the corresponding edge in the original graph. Edges with zero weight are shown in grey (weight value 0 not shown), (**II**) resulting MWPM, (**III**) resulting path of minimum weight (bold edges) which is obtained from the MWPM on $G_{A}$. For the example shown here, the path exhibits the weight $\omega_{p}=-1$. In addition, there are no additional loops in the graph but in principle they may be present.

**Figure 14 entropy-21-00193-f014:**
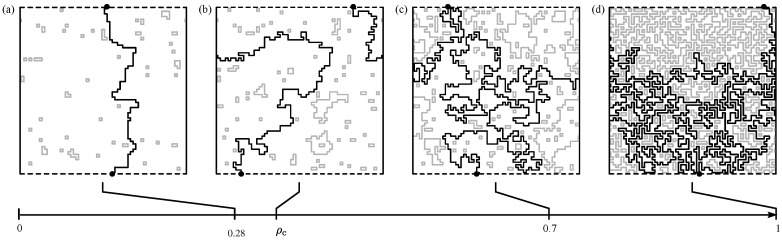
Examples for configuration which consist of a path of minimum weight plus loops. Here, a square lattice of size L=64 is shown. Dashed (solid) lines at the boundary represent free (periodic) boundary conditions. The MWP is shown in black. The selected nodes *s* and *t* are drawn with black dots. Gray lines are used to show the loops. The snapshots are taken at different values ρ for the disorder parameter; here (**a**) $\rho =0.28<\rho_{c};$ (**b**) $\rho \approx \rho_{c}=0.340\left(1\right)$; (**c**) ρ=0.7, and (**d**) full coverage ρ=1. The bottom scale indicates where the different configurations are located along the “disorder axis” in the range ρ=0…1.

**Figure 15 entropy-21-00193-f015:**
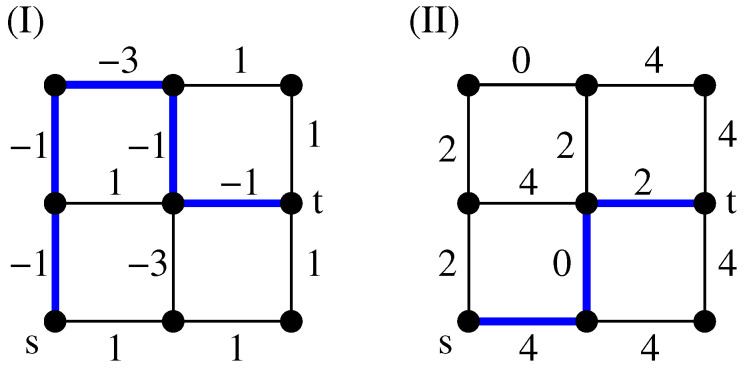
(**I**) Example for a minimum-weight s-t path (bold lines); (**II**) When shifting up all weight values, such that all weights are not negative, the shortest path changes.

**Figure 16 entropy-21-00193-f016:**
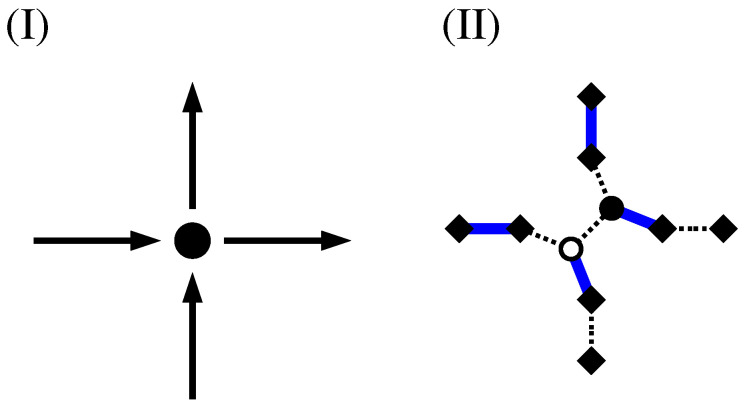
Illustration of the construction of the auxiliary graph $G_{A}$ for directed graphs *G*: (**I**) Node in the original graph with two incoming and two outgoing edges. (**II**) corresponding auxiliary graph with a sample perfect matching, representing a path using one ingoing and one outgoing edge.

**Figure 17 entropy-21-00193-f017:**
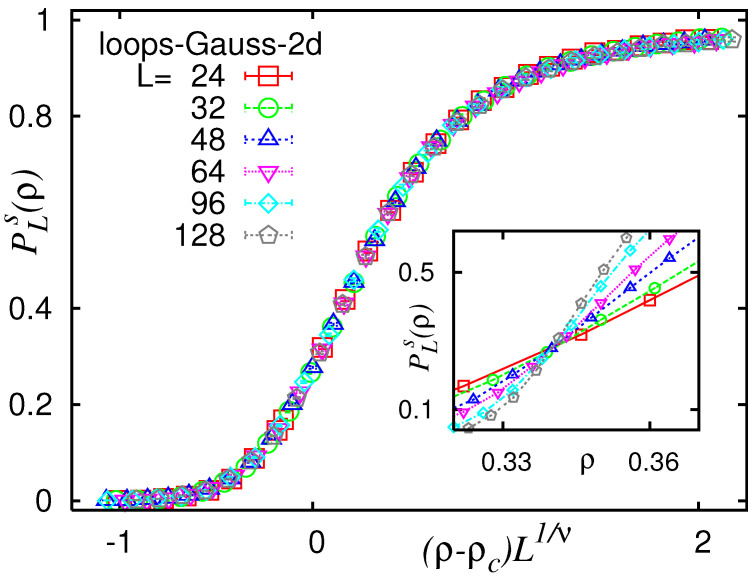
Probability of a percolating loop for two-dimensional lattices with mixed Gaussian disorder according to Equation (Equation 4) in Section 4.1. The inset shows the probability $P_{L}^{s}$ that the system exhibits a system spanning loop as a function of the disorder parameter ρ, for different system sizes *L*. The main plot shows the same data with the ρ-axis rescaled to determine the critical point $\rho_{c}$ and the critical correlation length exponent ν via a data collapse according to Equation (Equation 5) of this section.

**Figure 18 entropy-21-00193-f018:**
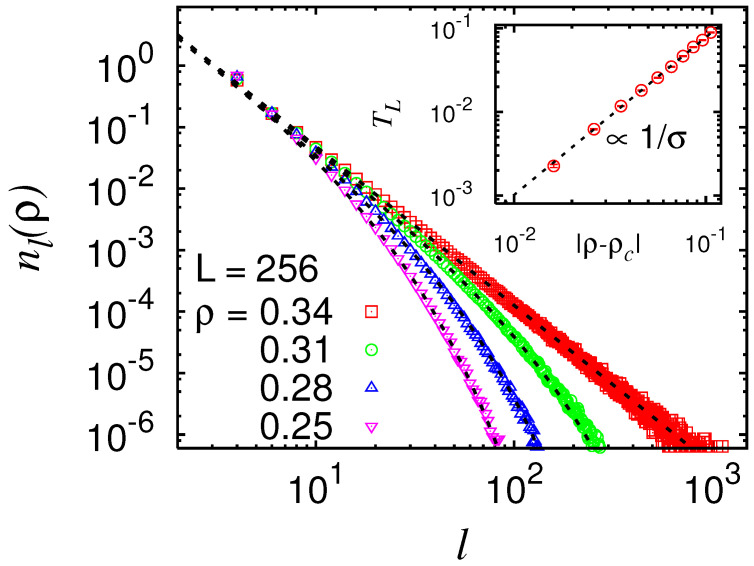
Distribution of the length of the non-percolating loops for two-dimensional NWP (L=256) with mixed Gaussian disorder according to Equation (Equation 4) for several values of the disorder parameter ρ. The lines show fits to the functions $n_{\ell }∼\ell^{-\tau }\exp\left(-T_{L}\ell \right)$. The inset shows the behaviour of $T_{L}$ as a function of the distance from the critical point.

**Figure 19 entropy-21-00193-f019:**
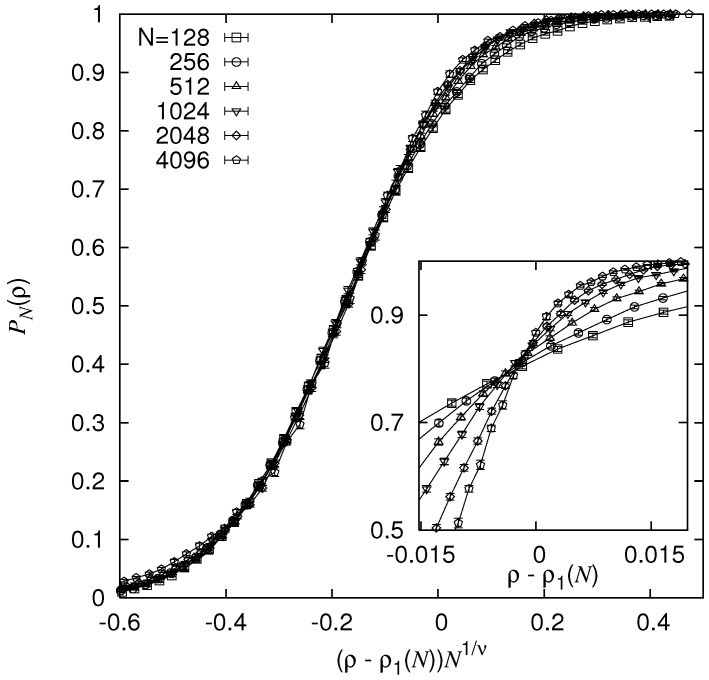
Probability that a path connecting two “furthest” nodes in the r=3-regular graph have a negative weight, corresponding to the fact that they would percolate if only negative paths would be allowed. The inset shows the raw data, renormalized by the size-dependent critical point $\rho_{1}\left(N\right)$. The main plot displays the collapse of data yielding the value of critical exponent ν.

**Figure 20 entropy-21-00193-f020:**
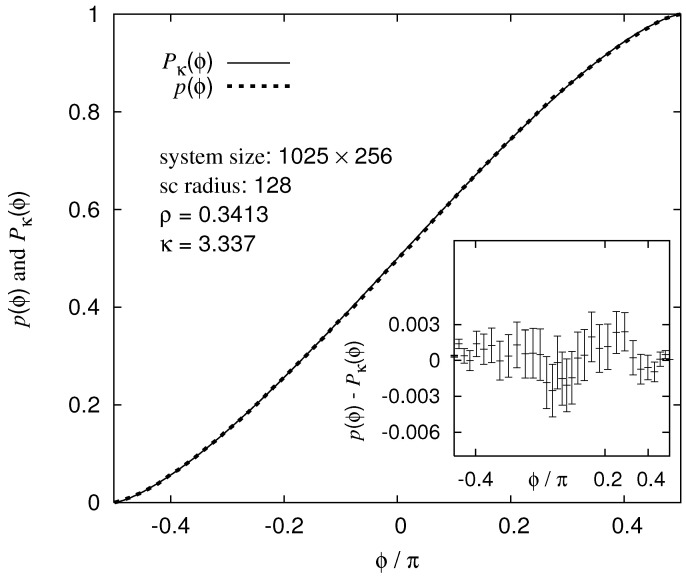
The left-passage probability p(ϕ) was obtained numerically for various points (257 different angles) on a semicircle (R=128). The simulations were performed right at the critical density for lattices with 1025 × 256 sites and averaged over 51,200 disorder realisations. The probability is shown together with the SLE prediction $P_{\kappa }\left(\phi \right)$. The diffusion constant $\kappa^{★}\;=\;3.343$ was chosen such that the agreement between $P_{\kappa }\left(\phi \right)$ and p(ϕ) is the largest. In the inset, the difference of the numerical left-passage probability to the prediction from SLE is shown, indicating that the agreement is indeed very good. To keep the inset clear, the difference is shown only for a selected number of typical data points.

**Figure 21 entropy-21-00193-f021:**
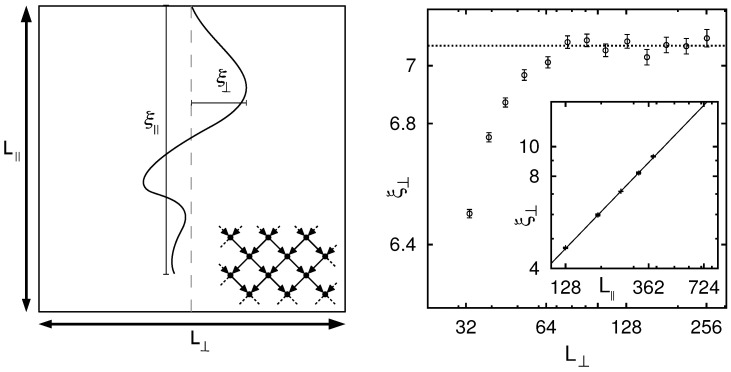
**Left**: determination of the correlations lengths $\xi_{∥}$ and $\xi_{⊥}$ which are parallel and perpendicular to the main (up to bottom) direction for directed NWP, respectively. **Right**: correlation length $\xi_{⊥}$ for different $L_{⊥}$ at fixed $L_{∥}=256$ to determine $\xi (L_{∥}=256,L_{⊥}\to \infty )$. The inset shows the scaling of $\xi (L_{∥},L_{⊥}\to \infty )$ to find the ratio $\nu_{∥}/\nu_{⊥}$.

**Table 1 entropy-21-00193-t001:** Critical points $\rho_{c}$ and critical exponents ν, β, γ, $d_{f}$, τ, σ for various dimensions *d*.

*d*	$\rho_{c}$	ν	β	γ	$d_{f}$	τ	σ
2	0.340(1)	1.49(7)	1.07(6)	0.77(7)	1.266(2)	2.59(3)	0.53(3)
3	0.1273(3)	1.00(2)	1.54(5)	−0.09(3)	1.459(3)	3.07(1)	0.71(1)
4	0.0640(2)	0.80(3)	1.91(11)	−0.66(5)	1.60(1)	3.55(2)	0.78(2)
5	0.0385(2)	0.66(2)	2.10(12)	−1.06(7)	1.75(3)	3.86(3)	0.88(2)
6	0.0265(2)	0.50(1)	1.92(6)	−0.99(3)	2.00(1)	4.00(2)	0.97(4)
7	0.0198(1)	0.48(1)	–	–	2.08(8)	4.50(1)	–

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
