# Peer review of "From Spin Glasses to Negative-Weight Percolation"

_entropy, 2019, doi:10.3390/e21020193_

Round 1

Reviewer 1 Report

This paper presents a very readable review of the authors’ work on negative-weight 

percolation, showing how it comes about in two-dimensional spin glass systems and then can be

generalized to any dimension (though not anymore related to the spin-glass problem, as I

understand it).  The authors explain their algorithm related to perfect matching; this algorithm

seems to be a big breakthrough in terms of solving this problem.  The negative-weight

percolation seems to have interesting properties in its own right, related to the loops formed.

The authors show that the percolation properties, especially the universality class, are different

from normal percolation, which indicates that there are long-range correlations built into

the system (the authors might mention this fact).  A good description of SLE and a 3-coodinated

random graph are also given.  I highly recommend publication.  I just have a few comments 

on the writing (I don’t know how much editing this journal does to the manuscripts):

1.  When using “which” in a sentence, it should refer to the subject right before it — otherwise

it is hard to read.  Thus suggest the sentence in the abstract be written, “Also, a summary 

is given for the results of NWP, which were obtained during the last decade”   or 

“Also, a summary of the results for NWP, which were obtained during the last decade, is given.”

There is a similar confusing “which” in the caption of Figure 4 — or maybe there one just

needs a comma after “positive energy.”

2.  After equation (1), no indentation.

3.  In a few places, the authors write “allows to” — this needs a subject, such as “allow one to”

or “allows us to”  

4.  p. 9   “super conductors”  (usually one word?)

5.  p. 10 “Anyway” —>  “In any case,”

6.  p. 12, line 350”  “on can” —>  “one can”

7. Critical exponents — table 1.  It is perhaps not surprising that they are different from percolation, 

as mentioned above with regards to correlations.  What about comparison with other systems — the 

Ising and Potts models, for example?  There the exponents are also known and are also different from percolation.

8.  Another way to look at critical systems is in the enclosed-area distribution.  Have the authors looked at that?  If not, it might be an interesting are for future research.

Author Response

Answers to points raise by referee 1:

1.  When using “which” in a sentence, it should refer to the subject
right before it — otherwise it is hard to read.  Thus suggest the
sentence in the abstract be written, “Also, a summary  is given for
the results of NWP, which were obtained during the last decade”   or
“Also, a summary of the results for NWP, which were obtained during
the last decade, is given.” There is a similar confusing “which” in
the caption of Figure 4 — or maybe there one just needs a comma after
“positive energy.”

-->
We have changed these sentences and checke all other occurences of "which",
which lead to a couple of additional changes.

2.  After equation (1), no indentation.

-->
changed

3.  In a few places, the authors write “allows to” — this needs a subject,
such as “allow one to” or “allows us to”  

-->
We have added subjects where they were missing

4.  p. 9   “super conductors”  (usually one word?)

-->
changed

5.  p. 10 “Anyway” —>  “In any case,”

-->
changed

6.  p. 12, line 350”  “on can” —>  “one can”

-->
changed

7. Critical exponents — table 1.  It is perhaps not surprising that
they are different from percolation,  as mentioned above with regards
to correlations.  What about comparison with other systems — the Ising and
Potts models, for example?  There the exponents are also known and
are also different from percolation.

-->
We now mention and compare also 2d Ising and Potts models (q=3,4) and the 2d
random bond Ising model.

8.  Another way to look at critical systems is in the enclosed-area
distribution.  Have the authors looked at that?  If not, it might be an
interesting are for future research.

-->
We have done this in Ref. [78]. We have added a corresponding sentence.

Reviewer 2 Report

This review of the negative weight percolation is well written. The problem is rooted in understanding spin glasses by studying domain walls. It leads to the negative weight percolation model. The authors argue in section 4. that their algorithms provide a more efficient way to study this percolation problem than the traditional Monte Carlo simulations.  I recommend publication. Below is a list of issues that if addressed can improve this article.

Statement on lines 277-278: “the studied objects …one dimensional, the loops may form fractals with df >1” seems self-contradictory. It needs clarifications.

In section 5.2 the mean field behavior is considered by using random graphs with 3 neighbors for each node. It should be interesting to relate this to a traditional mean field approach for percolation that uses the equivalent neighbor lattice (Erdos-Renyi network). See: M. Kaufman, M. Kardar, Phys Rev B 29, 5053 (1984).

The statement on line 42: “to see that a real thermodynamic phase transition takes place… one needs a divergence of some quantity” is incorrect. Critical phenomena are associated with any non-analytical behavior of the thermodynamic potential, e.g. models on Cayley trees M. Kardar, M. Kaufman, Phys. Rev. Lett. 51, 1210(1983).

In Section 3. The authors mention the “most fundamental” 2D random bond percolation problem. Connecting the problem of negative weight percolation to the classical bond percolation should improve this review article. Is there any way to relate the current model to the Kastaleyn-Fortuin (P. W. Kastaleyn and C. M. Fortuin, J. Phys. Soc. Jpn. (Suppl.) 26, 11 (1969)) expansion of Potts model?

Author Response

1.
Statement on lines 277-278: “the studied objects …one dimensional, the
loops may form fractals with df >1” seems self-contradictory. It needs
clarifications.

-->
We have removed the "one-dimensional" and just call the loops "line like",
while explaining that a line might fill a lattice to create
a fractal.

2.
In section 5.2 the mean field behavior is considered by using random
graphs with 3 neighbors for each node. It should be interesting to
relate this to a traditional mean field approach for percolation that
uses the equivalent neighbor lattice (Erdos-Renyi network). See:
M. Kaufman, M. Kardar, Phys Rev B 29, 5053 (1984).

--> This is an interesting suggestion, because also here one might find
continuously varying exponents. We have added a corresponding
remark and a citation to the given paper.

3.
The statement on line 42: “to see that a real thermodynamic phase
transition takes place… one needs a divergence of some quantity” is
incorrect. Critical phenomena are associated with any non-analytical
behavior of the thermodynamic potential, e.g. models on Cayley trees
M. Kardar, M. Kaufman, Phys. Rev. Lett. 51, 1210(1983).

-->
We have corrected this and state now only that the phase transition
is different from standard ones like for the ferromagnet.

4.
In Section 3. The authors mention the “most fundamental” 2D random
bond percolation problem. Connecting the problem of negative weight
percolation to the classical bond percolation should improve this
review article. Is there any way to relate the current model to the
Kastaleyn-Fortuin (P. W. Kastaleyn and C. M. Fortuin,
J. Phys. Soc. Jpn. (Suppl.) 26, 11 (1969)) expansion of Potts model?

-->
This is an interesting idea, but we do not see a direct application
in the moment: For FT, it is a local decision whether a bond
becomes "active" are not. For our NWP model on the other hand, it
depends on the states of many (potentially all) other bonds whether
a bond is part of a loop (part of a path) or not.